# Synthesis of Poly(methacrylic acid-co-butyl acrylate) Grafted onto Functionalized Carbon Nanotube Nanocomposites for Drug Delivery

**DOI:** 10.3390/polym13040533

**Published:** 2021-02-11

**Authors:** Josué A. Torres-Ávalos, Leonardo R. Cajero-Zul, Milton Vázquez-Lepe, Fernando A. López-Dellamary, Antonio Martínez-Richa, Karla A. Barrera-Rivera, Francisco López-Serrano, Sergio M. Nuño-Donlucas

**Affiliations:** 1Departamento de Ingeniería Química, Universidad de Guadalajara, Guadalajara 44100, Mexico; iq.josue.torres@gmail.com (J.A.T.-Á.); lcajero@yahoo.com.mx (L.R.C.-Z.); 2Departamento de Ingeniería de Proyectos, Universidad de Guadalajara, Zapopan 45100, Mexico; milton.vazquez@cucei.udg.mx; 3Departamento de Madera, Celulosa y Papel, Universidad de Guadalajara, Zapopan 45220, Mexico; ferdellam@gmail.com; 4Departamento de Química, Universidad de Guanajuato, Guanajuato 36050, Mexico; richa28@msn.com (A.M.-R.); fionita@ugto.mx (K.A.B.-R.); 5Departamento de Ingeniería Química, Universidad Nacional Autónoma de Mexico, Ciudad de Mexico 04510, Mexico; lopezserrano@unam.mx

**Keywords:** carbon nanotubes, nanocomposites, emulsion polymerization, methacrylic acid, nano-carriers, hydrogen bonds, hydrocortisone

## Abstract

Design of a smart drug delivery system is a topic of current interest. Under this perspective, polymer nanocomposites (PNs) of butyl acrylate (BA), methacrylic acid (MAA), and functionalized carbon nanotubes (CNTs_f_) were synthesized by in situ emulsion polymerization (IEP). Carbon nanotubes were synthesized by chemical vapor deposition (CVD) and purified with steam. Purified CNTs were analyzed by FE-SEM and HR-TEM. CNTs_f_ contain acyl chloride groups attached to their surface. Purified and functionalized CNTs were studied by FT-IR and Raman spectroscopies. The synthesized nanocomposites were studied by XPS, ^13^C-NMR, and DSC. Anhydride groups link CNTs_f_ to MAA–BA polymeric chains. The potentiality of the prepared nanocomposites, and of their pure polymer matrices to deliver hydrocortisone, was evaluated in vitro by UV–VIS spectroscopy. The relationship between the chemical structure of the synthesized nanocomposites, or their pure polymeric matrices, and their ability to release hydrocortisone was studied by FT-IR spectroscopy. The hydrocortisone release profile of some of the studied nanocomposites is driven by a change in the inter-associated to self-associated hydrogen bonds balance. The CNTs_f_ used to prepare the studied nanocomposites act as hydrocortisone reservoirs.

## 1. Introduction

In recent years, polymer nanocomposites (PNs) have encompassed several growing fields of research. PNs can be applied in multiple areas such as medicine, aerospace, packaging, electronics, agriculture, and the automotive industry, among others [1,2]. In contrast to composite materials, PNs are characterized by at least one of their phases having a size in the order of the nanometer range. Typically, this fact has had a positive influence on the outstanding properties that these materials exhibit.

Among other nanomaterials used as nanofillers to prepare PNs, carbon nanotubes (CNTs) are a desirable option due to their exceptional properties. Although CNTs can be used in medicine, optics, electronics, and other diverse fields, they show drawbacks (such as insolubility, potential toxicity, and complicated manipulation) that should be overcome before their full application to an industrial level [3]. In this sense, the preparation of CNT-based PNs is a simple way to surpass these inherent difficulties.

As a consequence of their large area, CNTs have a high aspect ratio (>1000), which favors their self-association. This fact complicates obtaining a homogeneous dispersion of CNTs in a polymer matrix. The challenge is not only to achieve the CNTs’ exfoliation, but it is also necessary to avoid a secondary CNTs agglomeration after they are added to the polymer used as PNs’ matrix. Several techniques have been developed in order to make the optimal dispersion of the CNTs possible [4].

Since it is necessary to prepare sufficient quantities of PNs (with a high grade of homogenization) for their commercial exploitation, there is a high level of interest to design a synthetic route that combines an optimal CNTs dispersion and a desirable amount of PNs produced. Chemical functionalization of CNTs, in situ polymerization, and emulsion polymerization are three techniques that can be used together in order to achieve this goal.

Chemical functionalization of CNTs is based on the attaching of a functional chemical group onto CNTs’ surface by covalent bonds. This is developed on sites of the CNTs prone to react, either at their end caps or on their sidewalls. The functionalization reactions carried out specifically in sp^3^-hybridized defects, pentagon–heptagon pairs (Stone–Wales defects), and vacancies in the CNTs lattice [5]. Among the known methods to graft polymers onto CNTs, the “grafting from” method stands out. This is a consequence of their easy control and availability for conjugating a wide type of polymers onto CNTs, such as poly(acrylates), polystyrene, or hyperbranched polymers, to mention a few [6].

In situ polymerization is an adequate technique for synthesizing structured high-performance CNT-based PNs. In this technique, a monomer is polymerized in the presence of CNTs. In some cases, the combination of functionalized CNTs and an in situ polymerization method have made it possible to prepare well-dispersed nanocomposites [7].

Emulsion polymerization is the industrial technique most often used to synthesize polymers and is also one of the available techniques for preparing CNT-based PNs. However, up to now, there are few references in the literature that focus on the preparation of nanocomposites using this technique [8,9,10]. CNT-based PNs prepared by emulsion polymerization usually have better mechanical properties, high UV resistance, strong interfacial forces, and a better surface appearance than the ones observed in polymers synthesized by the same polymerization technique [9]. In situ emulsion polymerization (IEP) is an interesting approach for synthesizing CNT-based PNs. This is because several issues associated with the nanocomposites synthesis can be addressed simultaneously. The main advantage of preparing PNs by IEP (where water is the continuous phase) is the presence of water surrounding every forming polymer particle. Hence, heat is easily dissipated, there is no viscosity increase, and very importantly, there is a suitable condition for CNTs’ exfoliation without causing any extensive thermal damage to polymer particles [10]. If a high grade of CNTs’ exfoliation is achieved, the interfacial attachment between the CNTs and the polymeric matrix chains can be improved [11]. The key issue to overcome the problems associated with the synthesis of CNT-based PNs by IEP is to minimize the tendency of the CNTs to agglomerate. As CNTs have a natural propensity to self-agglomerate forming physical entanglements, achieving homogeneous CNT-based PNs is a difficult challenge [12]. The most successful strategy to avoid the CNTs’ self-association by van der Waals forces is to achieve its chemical functionalization. This fact makes the synthesis of nanocomposites with functionalized CNTs by IEP very attractive from the chemical viewpoint.

To prepare CNT-based PNs with a specific chemical architecture following the “grafting from” strategy, the free-radical polymerization is a simple chemical route to exploit. An advantage of this technique is that it can be used to polymerize a large variety of monomers. Therefore, hydrophilic or hydrophobic polymers can be prepared by combining free-radical polymerization and IEP.

Poly(methacrylic acid) (PMAA) is a pH-sensitive polymer with a simple structure in contrast to other polymers, which have a complex structure and similar sensitivity [13]. The carboxylic acid groups contained in the methacrylic acid (MAA) units, in acidic conditions at pH lower than the p*K*_a_ (pH < 4), are no ionized. Therefore, these moieties are able to form hydrogen bonds inwardly in the bulk of this ionic polymer. As a consequence, PMAA forms a collapsed structure. However, as the medium pH rises, the PMAA structure changes. In the range of a pH between 5 and 7, carboxylic acid groups are deprotonated. When the pH reaches values in the alkaline region (pH > 7), PMAA chains attain an elongated rod-like conformation. These changes are reversible [14]. Therefore, PMAA is hydrophilic at pH 7.4 (typical pH of the human blood) and hydrophobic at low pH (around 1–2). Based on its properties, PMAA can be used to create an oral drug delivery system [15]. CNTs functionalized with PMAA can be used as interesting drug delivery nanocomposites sensitive to the pH of the surrounding medium. The protonated/deprotonated behavior of PMAA implies a strong change in polarity. This fact, along with the repulsive Coulomb interactions, is of high relevance. Due to the formation of deprotonated structure, if the PMAA is mixed or attached to a hydrophobic polymer, the preparation of a drug delivery system can be favored.

Butyl acrylate (BA) is a hydrophobic monomer and is used as a “soft monomer.” Some of its copolymers have been reported as biocompatible [16]. The low-temperature properties and toughness of copolymers prepared with BA can be improved, based on the soft nature of this monomer. The BA copolymers’ properties can be tailored to meet specific requirements.

Although the synthesis of the MAA–BA copolymer has been reported elsewhere [17], its practical applications have not been studied extensively. In this respect, its possible use as a drug delivery system is of special interest.

The development of new drug delivery systems seeks to enhance drug targeting. At the same time, improving drug efficiency and reducing the toxic effects is a desirable goal. For this, the design of drug transporters is a crucial factor, because they have a decisive influence on drug delivery and drug interaction [18]. CNTs grafted onto biopolymers have a high potential for use in the drug delivery field [19]. However, its possible toxicity is an unsolved problem. On the one hand, there are reports that point out that CNTs produce adverse effects on living tissues [20,21]. On the other hand, reports regarding the safe use of CNTs for humans have been presented elsewhere [22]; in particular, it is reported that functionalized CNTs have shown minor toxicity [23]. PMAA was the first polymethacrylate used in oral dosage form for drug delivery [24]. Two copolymers of PMAA have been applied with gastrointestinal (GI) tract targeting for drug delivery: (i) poly(methacrylic acid-co-ethyl acrylate) dissolved in duodenum at pH > 5.5, and (ii) poly(methacrylic acid-co-methyl methacrylate) dissolved in jejunum at pH > 6.0 or in colon at pH > 7.0 [25]. These copolymers are supplied under the trade name Eudragit for pharmaceutical applications. Broad release capabilities can be achieved if the ratio of monomers used to prepare MAA-based copolymers is changed.

Among other drugs, hydrocortisone (11β,17α,21-trihydroxypregn-4-ene-3,20-dione) is of particular interest due to its anti-inflammatory action. Hydrocortisone is an anti-inflammatory and hydrophobic drug that can be administered orally, topically, or by injection. This drug is widely used for the treatment of various diseases, such as certain types of allergies, arthritis, and asthma [26]. In addition, hydrocortisone has been used as treatment for several types of cancers, such as adrenocortical cancer [27], prostate cancer [28], leukemia [29], or lymphoma [30], among others.

The aim of this work is to prepare a pH-sensitivity novel drug delivery system synthesized from PNs with functionalized carbon nanotubes (CNTs_f_) as a reinforcing agent, and an MAA–BA copolymer as a polymeric matrix. To the best of our knowledge, the synthesis of the nanocomposites mentioned, following the chemical route presented here, has not been previously explored. The synthesis was carried out by IEP, under free-radical polymerization using water as continuous media, through a “grafting from” approach. In contrast to previously published in other works for drug delivery systems prepared with PMAA [24,25], in this work, BA co-monomer and functionalized CNTs were used in the preparation. It is expected that the last two components of the studied PNs increase their capacity to deliver drugs. The prepared nanocomposites contain a pH-sensitive copolymer (poly(MAA-*co*-BA)) attached to functionalized CNTs by a hydrolytically cleavable bond (anhydride groups). The morphology of the studied nanocomposites depends on the BA/MAA wt.% ratio, as well as the type and content of CNTs_f_ used. As a model for drug delivery study, their capacity to deliver hydrocortisone was evaluated. The presence of CNTs is a crucial factor in the ability of studied PNs to release hydrocortisone. The synthesized PNs could be used as smart nano-carriers with a targeted drugs’ release capast.

## 2. Experimental

### 2.1. Materials and Methods

Alumina boat was acquired from Alfa Aesar. Potassium persulfate (KPS) (reagent grade), hydrochloric acid ACS reagent 37%, and nitric acid ACS reagent 70% were purchased from Fermont México. Iron(III) nitrate nonahydrate Fe(NO_3_)_3_·9H_2_O > 98%, dichloromethane ACS reagent > 99%, acetone ACS reagent > 99%, methanol ACS reagent > 99%, sodium carbonate 0.1 N, hydrochloric acid 0.1 N, and sodium hydroxide 0.1 N were purchased from Golden Bell México. Oxalyl chloride 98%, triethylamine (Et_3_N) 99.5%, ethyl alcohol absolute 99.8%, potassium bromide FT-IR grade > 99%, tetrahydrofuran (THF) anhydrous > 99%, lithium hydride 95%, aluminum 99.9%, butyl acrylate > 99%, sodium dodecyl sulfate 98%, methacrylic acid 99%, and hydrocortisone 98% were purchased from Aldrich. Nitrogen gas and argon gas were acquired from Praxair México. Double-distilled water was purchased from “Productos Selectropura” México. Potassium biphthalate buffer pH 5 was purchased from Jalmek and used as received.

### 2.2. Synthesis and Purification of CNTs

CNTs were prepared via chemical vapor deposition (CVD) following an experimental procedure reported by our group elsewhere [31]. CNTs were purified with overheated steam. For this, 2 g of unpurified CNTs, wrapped in steel mesh, were placed in a quartz tube (1.3 cm i.d. and 25.4 cm long). Then, the overheated steam (at 873.15 K and 85.3 MPa) was fed for 3 h. This procedure was repeated twice. Purified CNTs were named CNTs_p_.

### 2.3. Functionalization of CNTs

CNTs_p_ were functionalized to insert methacrylic acid (MAA) molecules onto its surface before their use in the nanocomposites preparation, following an experimental procedure of several steps (the sequence is depicted in Scheme 1): (i) CNTs_p_ were partially oxidized to insert hydroxyl, carboxyl, and carbonyl groups onto their walls. The product, partially oxidized CNTs, was named CNTs_o_. (ii) To increase the amount of hydroxyl groups, a reduction of carboxyl groups was carried out through a treatment of CNTs_o_, with lithium aluminum hydride. The product was named CNTs_r_. (iii) CNTs_o_ and CNTs_r_ were functionalized separately to insert acyl chloride groups onto their walls. For this, both types of CNTs were treated with oxalyl chloride (OxCl). The products were named CNTs_o-OCl_ and CNTs_r-OCl_. (iv) CNTs_o-OCl_ and CNTs_r-OCl_ reacted separately with MAA molecules. Two types of products were obtained separately and named CNTs_o-OCl-MAA_ and CNTs_r-OCl-MAA_.

A brief description of the aforementioned chemical route is described next. Step 1: Three grams of the CNTs_p_ was placed into a 250 cm^3^ glass Soxhlet apparatus. Then, 150 cm^3^ of 8 M nitric acid solution was added. The dispersion was kept under reflux for 4 h. The obtained CNTs_o_ were purified by separating them from the liquid phase by centrifugation, washed several times with distilled water, and dried at room temperature until constant weight. Step 2: Due to the fact that OxCl preferentially reacts with hydroxyl groups, a certain amount of the CNTs_o_ carboxyl groups was reduced to hydroxyl groups. For this, 25 cm^3^ of anhydrous THF, 0.50 g of lithium aluminum hydride, and 0.7 g of CNTs_o_ were placed in a perfectly dry 50 cm^3^ flask. The reaction was carried out during 3 h at room temperature, under vigorous stirring. Next, 50 cm^3^ of methanol was slowly added. Then, 10 cm^3^ of a 0.1% HCl aqueous solution was added. Later, 10 cm^3^ of a 1% HCl aqueous solution was also added to achieve a complete neutralization of the original dispersion. The mixture was filtered out and the obtained CNTs_r_ were purified by repeated washing with distilled water and dried at room temperature until achieving constant weight. Step 3: A functionalization reaction between CNTs_o_ and CNTs_r_ with OxCl was carried out separately. For this, 0.1 g of CNTs_o_ or 0.1 g of CNTs_r_ was placed separately in a 200 cm^3^ flask, immersed in a water bath at 313.15 K. A mixture of 90 μdm^3^ of Et_3_N and 5 cm^3^ of dichloromethane was added with a syringe. The mixture was kept under agitation for 15 min. After this, a mixture of 60 μdm^3^ of OxCl and 5 cm^3^ of dichloromethane was added with a syringe, drop by drop. The reaction was carried out under agitation and N_2_ bubbling for 3 h. The product (CNTs_o-OCl_ or CNTs_r-OCl_) was used immediately after having been obtained. In order to analyze the obtained CNTs_o-OCl_ or CNTs_r-OCl_, a small amount was withdrawn from the flask and dried completely in a vacuum oven at 298.15 K for 24 h, to eliminate the residual mixture of OxCl and dichloromethane. Step 4: In the same flask where CNTs_o-OCl_ or CNTs_r-OCl_ was prepared, the necessary amount of MAA to produce the nanocomposites was added. The mixture was kept at room temperature under stirring for 30 min. Two types of products were obtained separately and named CNTs_o-OCl-MAA_ and CNTs_r-OCl-MAA_. Each type of product, as prepared, was used immediately to synthesize the nanocomposites, independently. To analyze the obtained CNTs_o-OCl-MAA_ and CNTs_r-OCl-MAA_, a sample of these products was withdrawn from the flask and heated for 1 h at 353.15 K to eliminate the excess of the MAA by evaporation.

The content of hydroxyl groups created onto CNTs_o_ and CNTs_r_ surface was determined by the indirect back titration method, based on the total acid groups titratable with NaOH. For this, an excess of OxCl was reacted with the CNTs_o_ and CNTs_r_ surface hydroxyl groups. Then, 0.1 g of CNTs_o-OCl_ or CNTs_r-OCl_ was mixed with 30 cm^3^ of NaOH 0.1 N. The mixture was reacted for 1 h. After this, the excess NaOH was determined by titration with a 0.1 N HCl solution.

The content of carboxyl groups grafted onto CNTs_o_ and onto CNTs_r_ surface was also determined by the indirect back titration method, using a weaker base (sodium bicarbonate). The method was previously reported [32]. Briefly, a specific amount of CNTs_o_ and CNTs_r_ was reacted with an excess of sodium bicarbonate. Carboxylic acid groups should react forming CNTs_o_–COO^−^Na^+^ and CNTs_r_–COO^−^Na^+^. The sodium bicarbonate excess was titrated with an HCl (0.1 N) solution.

### 2.4. Synthesis and Purification of CNTs_o-OCl-MAA_–BA and CNTs_r-OCl-MAA_–BA Nanocomposites and Their Pure Polymeric Matrix: MAA–BA Copolymer

The CNTs_o-OCl-MAA_–BA and CNTs_r-OCl-MAA_–BA nanocomposites were prepared by IEP. For this, immediately after CNTs_o-OCl-MAA_ and CNTs_r-OCl-MAA_ were prepared, the reagents necessary to prepare the nanocomposites were added in the same flask previously used. Specifically, a solution of 0.5 g of sodium dodecyl sulfate (SDS), dissolved in 95 g of water, was used. The mixture was stirred by immersion in an ultrasonic bath for 5 min. The flask was submerged in a water bath at 333.15 K and N_2_ gas was bubbled. After 30 min, the necessary amount of BA and a 0.1 g of KPS solution in 5 g of water were added. At this point, the reaction was started.

In order to perform an adequate comparison, MAA–BA copolymers were also synthesized. For this, a similar route to the one described to prepare the CNTs_o-OCl-MAA_–BA and CNTs_r-OCl-MAA_–BA nanocomposites was followed. Now, in a 200 cm^3^ glass flask, a solution of 0.5 g of SDS and 95 g of water was added. The flask was submerged in a water bath at 333.15 K and N_2_ gas was bubbled. After 30 min, the necessary mass of MAA, BA, and a solution of 0.1 g of KPS in 5 g of water was added, initiating the reaction.

Table 1 lists the contents of MAA and BA used to prepare the MAA–BA copolymers, as well as of these co-monomers and CNTs_o_ or CNTs_r_ to prepare CNTs_o-OCl-MAA_–BA or CNTs_r-OCl-MAA_–BA nanocomposites.

Scheme 1 shows the chemical route used to prepare CNTs_r-OCl-MAA_–BA nanocomposites. A similar chemical path was used to prepare CNTs_o-OCl-MAA_–BA nanocomposites, but without the step that produces the reduction of carboxyl groups.

After synthesizing the CNTs_o-OCl-MAA_–BA and CNTs_r-OCl-MAA_–BA nanocomposites and the MAA–BA copolymers, these materials were purified. For this, the obtained latex was frozen to obtain a solid material. To dissolve the solid latex, a volume of acetone equivalent to the volume of latex was added. Then, the mixture was allowed to thaw-out. A precipitate formation was produced. The solid–liquid mixture was filtered, washed, and water-dialyzed at 343.15 K for 72 h to eliminate the residual SDS. The pH of the washing water was close to 7. Finally, the purified product was dried in an oven at 323.15 K until achieving a constant weight. The purified samples of the MAA–BA copolymers and of the CNTs_o-OCl-MAA_–BA and CNTs_r-OCl-MAA_–BA nanocomposites were studied by FT-IR, XPS, NMR, DSC, and UV–VIS techniques.

### 2.5. Characterization of MAA–BA Copolymers and Their CNTs_o-OCl-MAA_–BA and CNTs_r-OCl-MAA_–BA Nanocomposites

A battery of techniques was used to characterize synthesized and functionalized CNTs, MAA–BA copolymers, and their nanocomposites. There is special interest in verifying whether the functionalized CNTs were bonded chemically to the polymeric macromolecules of nanocomposites studied.

CNTs were examined with a field emission scanning electron microscope (FE-SEM), model MIRA 3LU of Tescan (Brno, Czech Republic). Samples were prepared by mixing ca. 0.02 g of CNTs with 2 cm^3^ of acetone, at room temperature. An aliquot of this dispersion was taken with a Pasteur pipette and poured onto a Cu grid. After solvent evaporation, the samples were analyzed in the FE-SEM microscope.

CNTs were also examined with a high-resolution JEOL (Tokyo, Japan) transmission electron microscope (model 2010 HRTEM operated at 200 kV). Here, the samples were prepared by mixing ca. 0.007 g of CNTs with 2 cm^3^ of acetone, at room temperature. The obtained mixture was sonicated for 5 min. After this, an aliquot of this dispersion was poured onto a Cu grid, using a Pasteur pipette. With a 60 W solar lamp, the solvent was evaporated for 15 min. After this, the sample was placed into the HR-TEM microscope for analysis.

Raman spectroscopy was used to analyze samples of functionalized CNTs. For this, a Dilor (Villeneuve-d’Ascq, France) spectrometer model Lab Raman II was used. This device is equipped with a HeNe laser, which was operated at 20 mW and at an excitation wavelength of 632.8 nm. The measurements were made using a 50× objective with an area spot of 2 mm and 2 cm^−1^ errors.

Structural characterization of CNTs_p_, CNTs_f_, CNTs_o-OCl-MAA_–BA and CNTs_r-OCl-MAA_–BA nanocomposites, and MAA–BA copolymers was performed by Fourier-transform infrared spectroscopy (FT-IR spectrophotometer, Spectrum One of Perkin Elmer, Waltham, Massachusetts, USA). For this, pellets formed with KBr and dry samples were prepared (at a ratio of 1 mg of the sample to 300 mg of KBr) by compression at room temperature. Reported spectra were obtained and analyzed from an average of 64 scans, to reduce the signal/noise ratio, and a resolution of 4 cm^−1^.

Nanocomposites of CNTs_o-OCl-MAA_–BA and CNTs_r-OCl-MAA_–BA and their pure copolymer matrix were analyzed by the X-ray photoelectron spectroscopy (XPS) technique. XPS was carried out using an XR50 M monochromatic Al Kα_1_ (*hν* = 1486.7 eV) X-ray source and a Phoibos 150 spectrometer. This equipment has a one-dimensional detector 1D-DLD, provided by SPECS (Berlin, Germany). A set of experiments was mounted on a steel sample holder using double-sided copper tape. Prior to the measurements, the samples were dried in a vacuum oven, at a temperature of 333.15 K for 48 h. After this, they were introduced into the pre-chamber. All measurements were recorded at 150 W and takeoff angle of 90°, setting the pass energy at 10 eV and a step size of 0.1 eV. The flood gun device was used for charge neutralization on samples. Moreover, spectra were shifted, using the C–C position on binding energy.

Solution ^1^H spectra in DMSO-d_6_ of the pure MAA–BA copolymers were recorded at room temperature on a Bruker (Billerica, Massachusetts, USA) Avance 500 spectrometer using tetramethylsilane (TMS) as an internal standard.

Solid-state ^13^C NMR spectra, of the nanocomposites, and of their pure copolymer matrices, were recorded under proton decoupling on a Bruker (Billerica, Massachusetts, USA) Avance 400, operating at 100.613 MHz for ^13^C. A Bruker probe equipped with 4 mm rotors was used. CP-MAS spectra were obtained under Hartmann-Hahn matching conditions and a spinning rate of 6.0 kHz was used. A contact time of 2.5 ms and a repetition time of 4 s were used. The measurements were made using spin-lock power in radiofrequency units of 60 kHz and 4000 transients were typically recorded. Chemical shifts were externally referenced to tetramethylsilane (TMS), using adamantane.

DSC thermograms of the prepared nanocomposites and of their pure polymer matrices were obtained on a TA-Instruments (New Castle, Delaware, USA) calorimeter model Q-100. DSC thermograms were recorded following a heating program from 183.15 to 453.15 K at a heating rate of 10 K/min, using a flow rate of 50 cm^3^/min of nitrogen to maintain an inert atmosphere. Two scans were carried out and the second scan is reported.

A study of hydrocortisone release was carried out in vitro by UV–VIS spectroscopy. For this, tablets containing hydrocortisone were prepared previously. To prepare the tablets, samples of the nanocomposites or of their pure polymer matrix were mixed with 1% (*w*/*w*) of hydrocortisone at room temperature. Then, they were placed in a 7 mm diameter mold and pressed to obtain tablets with 1.5 mm of thickness. Tablets’ final weights were in the range of 35 to 55 mg. After tablets preparation, they were placed in 3 cm^3^ of potassium biphthalate buffer at pH 5 in a quartz cell. The hydrocortisone concentration was determined by the variation of absorbance as a function of time on a Perkin-Elmer (Waltham, Massachusetts, USA) UV–VIS spectrometer Lambda 25, at 248 nm. In addition, the swelling degree of the nanocomposites, and of their polymeric matrix, was measured at room temperature. This was carried out by preparing tablets of pure nanocomposites or MAA–BA copolymers with the same size and weights as those already described above. All tablets were dried until observing constant weight. Then, they were place in a vial with 10 cm^3^ of potassium biphthalate buffer at pH 5 for 24 h. After this, at certain times, the swollen tablets were then taken out of the vial, wiped with a filter paper, and weighed. The swelling degree (*S_w_*) was calculated with the following equation:(1)$$S_{w}=\frac{W_{w}-W_{d}}{W_{d}}$$
where *W_d_* is the weight of dried tablet and *W_w_* is the weight of the swollen tablet.

## 3. Results and Discussion

Figure 1A depicts a CNTs_p_ micrograph, obtained by FE-SEM. It is evident that the population of CNTs_p_ is formed mainly by helical carbon nanotubes (HCNTs). The length of some of them exceeds 2 μm. Due to the microscope resolution capability, superficial imperfections cannot be detected. The HCNTs’ coil-shaped geometry favors their mechanical entanglement with a polymeric matrix. However, to assure a homogeneous dispersion of these nanostructures, it is highly recommended that their surface be previously functionalized. Due to the helical configuration of HCNTs, in which pentagonal/heptagonal carbon rings are periodically repeated, forming superficial defects, the chemical HCNTs’ functionalization differs from that used for neat CNTs [33]. Therefore, HCNTs’ functionalization requires suitable chemical conditions [34]. In this sense, the chemical modification method to be used can take advantage of the high surface energy state of the HCNTs, induced by their tensile strength, caused by their helical structure [35].

Figure 1B shows an HR-TEM micrograph of the CNTs_p_, including a 20 nm bar scale, where multiple walls for the structure of CNTs_p_ can be observed. Based on this micrograph, it is evident that the synthesized and purified CNTs here are helical multiwalled carbon nanotubes (HMWCNTs). In addition, it can be observed that 39 carbon sheets form the walls of the CNTs_p_ sample analyzed. Their inner diameter is 13.8 nm, while the outer diameter is 41.6 nm. The synthesis of HMWCNTs following a catalytic chemical vapor deposition method with Fe nanoparticles as catalysts was reported previously. Nonetheless, in this work, we use a different experimental approach to achieve the HMWCNTs’ synthesis, regarding a previously reported approach [36]. HMWCNTs have interesting properties; for example, they can act as an efficient material for the dispersive solid-phase extraction of low and high molecular weight polycyclic aromatic hydrocarbons, from aqueous solutions [37]. In this work, we report the use of synthesized HMWCNTs to prepare CNTs_o-OCl-MAA_–BA and CNTs_r-OCl-MAA_–BA PNs nanocomposites.

The functionalization of prepared CNTs was assessed by various techniques. Initially, the presence of hydroxyl and carboxyl groups on CNTs’ walls was evaluated. The amount of hydroxyl and carboxyl groups was quantified following the procedure previously described in the Experimental section. It was found that the amount of hydroxyl groups was 2.3 mmol per gram in CNTs_o_ and 7.6 mmol per gram in CNTs_r_, while the amount of carboxyl groups was 2.8 mmol per gram of CNTs_o_ and 1.3 mmol per gram in CNTs_r_. These results confirm that the transformation of carboxyl groups attached to CNTs_o_ walls to hydroxyl groups inserted on CNTs_r_ walls was successfully achieved.

Figure 2 shows the functionalized CNTs’ Raman spectra. The four Raman spectra show a similar pattern: three peaks are clearly resolved. In the CNTs_r_ spectrum, other Raman contributions of minor intensity are detected. Table 2 lists the frequencies of the three mentioned bands. The graphite band (G-band) is detected at ca. 1574 cm^−1^. This is a first-order Raman band, and is related to the tangential vibration of two carbon atoms, in one graphene unit cell. This cell is the basic structure of graphene sheets that rolled-up to form the CNTs’ walls. G-band is associated with the crystallinity degree of graphene sheets. In addition, it is considered to be due to the highly ordered carbon structures, like those that form the nanotube walls. As a shoulder of the G-band, the G*-band, at around 1612 cm^−1^, was detected. The G*-band is a lesser-known disorder-induced band [38]. The results listed in Table 2 reveal that the G-band of HMWCNTs, functionalized with acyl chloride functionality (CNTs_o-OCl_ and CNTs_r-OCl_), shifts at lower frequencies, with respect to their precursors (CNTs_o_ and CNTs_r_). This is because the CNTs’ vibrational response is modified by the introduction of guest molecules, with donor or acceptor electronic characteristics [39,40]. This behavior confirms that the chemical groups attached to the walls of CNTs_o-OCl_ and CNTs_r-OCl_ have a different chemical structure, as compared to those attached to CNTs_o_ and CNTs_r_. At lower frequencies, a disorder-induced mode, called D-band (at ca. 1323 cm^−1^), was detected. This band originates from a double-resonance process, in which an elastic scattering and an inelastic scattering caused by a defect and by a phonon, respectively, is developed [41]. In a fashion similar to the one observed for the G-band, the D-band on the samples of the CNTs_o-OCl_ and CNTs_r-OCl_ was detected at a lower frequency, with respect to that of their CNTs_o_ and CNTs_r_ precursors. Table 2 also lists two ratios, calculated from integrated areas of the D and G bands (I_D_/I_G_) or the G* and G bands (I_G*_/I_G_). These ratios are typically used to evaluate the disorder degree on CNTs’ walls. This disorder is related to the amount of functional groups covalently attached to the CNTs’ surface. It is evident that both ratios increase for CNTs_o-OCl_ with respect to CNTs_o_, but decrease for the CNTs_r-OCl_ compared to CNTs_r_. This result indicates that a reordering within the carbon nanotubes CNTs_r-OCl_ has taken place. As the amount of hydroxyl groups in CNTs_r_ is larger than that in CNTs_o_, it is likely that the hydrogen bonds of carbon nanotubes of CNTs_r-OCl_ increase with respect to the ones present in CNTs_o-OCl_. This factor would influence the creation of a more ordered assembly of carbon nanotubes in CNTs_r-OCl_. At higher frequencies, a D-band overtone called “G’-band” was observed. The G’-band is caused by a double-resonance process. The G’-band, for the different types of functionalized CNTs studied in this work, shows a different intensity. This is caused by the different structural arrangements of CNTs.

Figure 3 shows the FT-IR spectra of purified and functionalized CNTs. For comparison, the FT-IR spectrum of pure MAA monomer was included (Figure 3D). The CNTs_p_ FT-IR spectrum (Figure 3A) shows an intense wide band at 3430 cm^−1^, due to the hydroxyl groups’ stretching vibration, and a weak band at 798 cm^−1^, caused by out-of-plane bending vibration of the C–H bond. In the FT-IR spectrum of CNTs_o_ (Figure 3B), other spectral contributions appear. At 1020 cm^−1^, a band due to stretching vibration of the C–O bond was detected. The weak and wide band detected at 612 cm^−1^ is assigned to vibrations of the C–OH torsional band. Moreover, it is evident that the band, due to hydroxyl groups’ stretching vibration, becomes wider than the one detected in the CNTs_p_ spectrum (Figure 3A). This fact provides evidence that the population of hydroxyl groups increases in the CNTs_o_. The FT-IR spectrum of CNTs_o-OCl_ (Figure 3C) shows spectral contributions that indicate acyl chloride groups are attached to the CNTs_o-OCl_’ walls. Thus, the stretching vibration of the C–Cl bond appears at 807 cm^−1^, and the harmonic band of this vibration was detected at 1480 cm^−1^. The last band appears as a shoulder of the band detected at 1475 cm^−1^. This last peak is assigned to C–H bending vibration bonds, created near holes and imperfections in CNTs’ walls, due to the functionalization process. At 1764 cm^−1^, a weak band appears due to the stretching vibration of the carbonyl group that forms part of acyl chloride functionality. At 1037 cm^−1^, the band with a stronger intensity of this spectrum emerges. This band is assigned to vibration stretching of the C–O functionality of an ether group, which includes an aryl group. Due to the fact that CNTs have walls, typically formed by hexagonal carbon rings, they have an aromatic nature like that of the aryl group; therefore, this band confirms that the C–O functionality is attached to CNTs_o-OCl_ walls. Furthermore, the hydroxyl groups’ band detected at ca. 3500 cm^−1^ practically disappears (when compared with the observed in Figure 3A,B). This fact strongly suggests that the hydroxyl groups of the precursor CNTs_o_ have reacted. The FT-IR spectrum of the pure monomer MAA (Figure 3D) shows that their carboxyl groups are grouped forming dimers. This is because at 1701 cm^−1^, a band appears due to the stretching vibration of carboxyl groups self-associated by hydrogen bonding. Besides, the stretching of the C=C bond produces the band observed at 1634 cm^−1^. Regarding the CNTs_o-OCl-MAA_ FT-IR spectrum (Figure 3E), it was analyzed whether a chemical reaction between MAA and CNTs_o-OCl_ was performed. In this spectrum, a weak band at 1848 cm^−1^ appears. This band is due to the symmetric stretching vibration of the anhydride group. In addition, the antisymmetric stretching vibration of the anhydride group was detected by another weak band, appearing at 1785 cm^−1^ as a shoulder of the 1740 cm^−1^ band. The last signal is assigned to the stretching vibration of a carbonyl group of ketone functionality. The detection of IR bands characteristic of vibrations of anhydride groups indicates that a grafting reaction did occur between the superficial acyl chloride groups of CNTs_o-OCl_ and the MAA hydroxyl groups. The stretching vibration of the C–O bond of an ether functionality causes the band detected at 1021 cm^−1^, while the stretching vibrations of C–H bonds of methyl groups and hydroxyl groups produce the intense and broad signal, detected at 2991 cm^−1^ [42]. The last groups are associated via hydrogen bonds with carbonyl groups. The spectral behavior described above confirms that MAA molecules reacted with acyl chloride groups, present in the walls of CNTs_o-OCl_, to form the product that we called CNTs_o-OCl-MAA_. Similar behavior was observed in the CNTs_r-OCl-MAA_ FT-IR spectrum.

^1^H spectra of Mat 1 and Mat 2 were obtained (shown in the Appendix A). In these spectra, the typical signals of MAA and BA co-monomers were identified. Both co-monomers did react to form the pure polymer matrix of the nanocomposites studied. The spectral contributions observed in these spectra confirm that the MAA–BA copolymer was synthesized successfully following the chemical route reported here. Unfortunately, overlapping of the signals of MAA and BA prevents calculating the composition of MAA–BA copolymer.

Figure 4 depicts the ^13^C NMR CP-MAS spectra of Mat 2 (Figure 4A) and Nano 6 (Figure 4B). The chemical structure of the MAA–BA copolymer has been inserted in Figure 4A. The following peaks can be observed: at 173.8 ppm, a carbonyl group signal *a*; at 65.6 ppm, a peak (identified as signal *g*) of the methylene carbon of the O=C–O–CH_2_– ester functionality [43]. Further, at 42.8 ppm, the signal *f* of the methylene groups of the BA units and signal *c* for the MAA moiety can be observed, as well as the signal *e* at 33.3 ppm corresponding to –CH–C=O functionality. The signal *h* at 29.4 ppm corresponds to the carbon of methylene groups of the –CH_2_–CH_2_– functionality. Finally, at 26.0 ppm, the signals *j*, *i* of the carbon of methylene and methyl groups (–CH_2_–CH_3_ functionalities) of BA units were detected. The methyl group, signal *d*, can contribute to the intensity of the signal detected at 26.0 ppm. These signals confirm the synthesis of an MAA–BA copolymer.

The ^13^C NMR CP-MAS spectra shown in Figure 4 are alike. The only differences emerge in the aliphatic carbons’ sidebands that appear at 48 and 10 ppm in the Nano 6 spectrum, which were absent in the Mat 2 spectrum. Both samples were rotated at the same rate. In a CP-MAS experiment, the purpose of rotation about the magic angle is the isotropic average of the chemical shift anisotropy (CSA) shielding tensor to a single peak [44,45]. The appearance of spinning sidebands is because the magic angle spinning (MAS) rate is less than the CSA frequency range. It is expected that the presence of the nanofiller (CNTs_r-OCl-MAA_) induces a different pattern in the sidebands of the nanocomposite. This is due to the fact that nanofillers induce the formation of ordered domains in their vicinity, not present in the amorphous MMA–BA copolymer. In the ordered domains, chemical shift anisotropy of polymeric backbone cannot be completely eliminated at the spinning rate used.

Figure 5 shows the normalized C1s core level spectra of Mat 2 (Figure 5A), Nano 6 (Figure 5B), and Nano 8 (Figure 5C). To fit the obtained data, we used the active background approach, which is accomplished by the software AAnalyzer 1.42 [46]. All spectra were adjusted to their respective shifts, according to the C–C/C–H bonds position at 284.8 eV. The normalized C1s core level spectrum of Mat 2 (Figure 5A) shows four components: at 284.8 eV, a contribution due to C–C/C–H bonds [47], at 285.5 eV this contribution is assigned to C–C=O functionality [48], the contribution observed at 286.4 eV is due to ether (C–O) bond [47,48], while the contribution that appears at 288.9 eV is due to ester (O=C–O) functionality [47,48]. The calculated atomic concentrations of these groups are 33.7%, 17.7%, 9.0%, and 8.2%, respectively. In the normalized C1s core level spectra of the Nano 6 (Figure 5C) and Nano 8 (Figure 5B), the same contributions were found at the same positions. Specifically, for the Nano 6, the components and their atomic concentrations are 284.8 eV (C–C/C–H) 36.3%, 285.4 eV (C–C=O) 20.7%, 286.4 eV (C–O) 5.1%, and 288.9 eV (O=C–O) 12.7%. For Nano 8, the components and their atomic concentrations are 284.8 eV (C–C/C–H) 33.6%, 285.4 eV (C–C=O) 22.3%, 286.4 eV (C–O) 10.1%, and 288.9 eV (O=C–O) 10.3%. For both nanocomposites, the more significant change was that the atomic concentration of functionalities containing carbonyl group increases. In particular, this highlights the changes detected corresponding to the O=C–O functionality contribution. Thus, for Mat 2, the concentration of this functionality was 8.2%, while Nano 6 was 12.7% and Nano 8 was 10.3%. For the anhydride functionality (R_1_–(C=O)–O–(C=O)–R_2_), it has been reported that the signal for the anhydride group is observed at 289.42 or 289.36 eV [48]. The integration for the contribution mentioned was made from 287.83 to 290.23 eV. Therefore, the change observed in the XPS spectra of the nanocomposites Nano 6 and Nano 8 indicates that anhydride groups were created as a consequence of the chemical route used for their preparation.

Table 3 lists the areas of two types of signals: (i) carbonyl bond found by ^13^C-NMR and (ii) O–C=O functionality detected by XPS. The ratio of the carbonyl group to methyl group areas (the stronger signal of ^13^C NMR CP-MAS spectra) is reported. In a similar way, the ratio of the signals of O–C=O functionality to C–H bonds (the stronger signal of XPS spectra) was also recorded. Consequently, both ratios are normalized signals. It is evident that there is a close similarity in these ratios. Although the levels of analysis for XPS (over the surface) and NMR (the complete mass of the sample) techniques are different, these results show that more carbonyl groups were created in the nanocomposites, with respect to those detected on pure MAA–BA copolymers. Therefore, because O–C=O functionality is part of the anhydride groups, it is evident that these results support the previous FT-IR analysis. The anhydride groups are the result of the reaction between acyl chloride groups, attached to CNTs’ wall, and hydroxyl groups of MAA monomer. The chemical reaction between acyl chloride groups and hydroxyl groups, to form anhydride groups, has been previously documented for other systems [49]. The formation of MAA–BA chains was carried out from CNTs’ walls, where hydroxyl groups of MAA reacted with acyl chloride groups attached to CNTs_o-OCl_’ walls or to CNTs_r-OCl_’ walls, forming anhydride groups in CNTs_o-OCl-MAA_ or in CNTs_r-OCl-MAA_, and allowing a C=C bond being available to polymerize by radical polymerization under IEP process. Therefore, it can be stated that MAA–BA chains were grafted onto the surface of the functionalized CNTs. It is expected that the grafting efficiency of this reaction is not total.

Figure 6 shows DSC thermograms of the pure MAA–BA copolymer matrix and the ones of the nanocomposites prepared with 0.5 wt.% of CNTs_o_ or CNTs_r_. Table 4 shows the thermal transition detected on the corresponding thermograms. This thermal transition is a glass transition temperature. Both poly(methacrylic acid) (PMAA) and poly(butyl acrylate) (PBA) are amorphous polymers. The reported glass transition temperature (*T*_g_) of PMAA is 501.15 K [50], while the *T*_g_ of PBA is 221.15 K [51]. The Mat 1 and Mat 2 DSC thermograms show only one *T*_g_ in each case. The *T*_g_ of Mat 1 is 6 279.15 K, while the *T*_g_ of Mat 2 is 304.15 K. As expected, as more MAA monomer is used to prepare the MAA–BA copolymer, the *T*_g_ of MAA–BA copolymer is detected at higher temperatures. The fact that the amorphous phase of Mat 1 and Mat 2 has only one thermal relaxation (one *T*_g_) indicates that both copolymers have a random structure, which is the typical structure of copolymers, prepared following a free-radical polymerization. For the nanocomposites, one or two thermal transitions were detected: (i) in the DSC thermograms of Nano 1, Nano 5, Nano 6, and Nano 8, only one *T*_g_ was detected, while (ii) the DSC thermograms of Nano 2, Nano 3, Nano 4, and Nano 7 nanocomposites show two *T*_g_s. Since in general, for nanocomposites prepared with a high amount of BA and a low amount of CNTs_o-OCl_ or CNTs_r-OCl_, two glass transition temperatures are observed, we consider that both factors facilitate the segregation of BA and MAA domains, which, in turn, is confirmed by the two observed *T*_g_s. In this respect, a low amount of CNTs_o_ or CNTs_r_ favors the homogeneous dispersion of both types of CNTs_f_. In addition, a higher amount of BA favors that, even though the polymerization process is carried out by free radicals, domains rich in BA are present in the MAA–BA chains. This statement can be supported by the results reported by Kulikov et al. [52], which show that the reactivity of MAA in emulsion polymerization is higher, by a factor of at least 4, than the reactivity of BAA. In emulsion copolymerization, the initiation can be carried out in the aqueous phase, or inside the micelles. Thus, the solubility and concentration of the co-monomers play a key factor in defining the locus where initiation reactions are mainly developed. Since MAA is highly soluble in water, it is very probable that at low reaction times, MAA oligomers created in the aqueous phase react with CNTs_o-OCl-MAA_ or CNTs_r-OCl-MAA_. However, as the reaction time proceeds, the BA units react preferentially. This would explain how MAA-rich domains and BA-rich domains were formed. A homogeneous dispersion of CNTs_o-OCl-MAA_ or CNTs_r-OCl-MAA_ aids in carrying out this process performance. On the contrary, when the co-monomeric formulation is richer in MAA, both MAA and BA molecules react during the entire IEP process. In this situation, the formation of segregation domains rich in BA or MAA is more difficult. As a consequence, a random structure is formed in MAA–BA chains, which was detected by the appearance of a single glass transition temperature, as was detected in the DSC thermograms of Nano 1, Nano 5, Nano 6, and Nano 8.

The morphology of the studied nanocomposites has crucial importance in their possible use as a drug delivery system sensitive to the medium pH. In this sense, a particular drug can be preferentially adsorbed in their hydrophilic or hydrophobic domains and, as has been reported for other stimuli-responsive polymers, used as smart nano-carriers [53]. The pH-responsive behavior of the MAA would facilitate the controlled delivery of drugs. This is highly desirable because different tissues and cells have varied pH values [54].

Figure 7 shows the hydrocortisone release profiles for pure MAA–BA copolymer Mat 2 and for the nanocomposites Nano 5 and Nano 6 obtained at pH 5. These nanocomposites, and their pure polymer matrix, were prepared with the same initial content of MAA (40 wt.%) and in the same way, they have a random chemical structure. It is evident that the amount of hydrocortisone released from Nano 6 is higher than those observed for the other two materials. At 73 h, Nano 6 releases 23% of hydrocortisone. These results contrast with the ones observed for Mat 2 and Nano 5. In this regard, the poor hydrocortisone release from Mat 2 indicates that this MAA–BA copolymer has little capability to act as a hydrocortisone carrier. These results mean that the type of CNTs_f_ used to prepare the nanocomposites (CNTs_o_ for Nano 5 or CNTs_r_ for Nano 6) has a crucial effect on the behavior of hydrocortisone release observed. In this regard, Table 3 shows that the ratios of C=O/CH_3_ signals, measured by ^13^C-NMR and O–C=O/C–H signals, evaluated by XPS, are higher for Nano 6 than Nano 5. These results also suggest that the content of CNTs_r_ in Nano 6 is higher than the CNTs_o_ content in Nano 5. Therefore, there is a direct relationship between the hydrocortisone release capability and the content of CNTs_f_ in the polymer matrix of the nanocomposites studied. Our results confirm that it is possible to fill the internal cavity of CNTs with a drug to create drug reservoirs, as reported by Hampel et al., who used carboplatin as a filling drug of oxidized MWCNTs [55]. At pH 5, the collapsed structure of PMAA starts to change because their carboxylic groups (completely self-associated by hydrogen bonds at pH < 4) now can be deprotonated. A change in the balance from inter-associated to self-associate hydrogen bonds can be the driving force to the release of hydrocortisone charged into the CNTs_f_. With respect to the swelling degree of the nanocomposites and of their polymer matrix, after 48 h of the swell-test at pH 5, the swelling could not be calculated. Two factors explain these results: (i) the nanocomposites and their polymeric matrix are mainly hydrophobic, and (ii) at pH 5, the MAA units inserted in the two materials mentioned are self-associated by hydrogen bonding. This means that there is no interaction with water and, as a consequence, no swelling was detected.

Figure 8 and Figure 9 show FT-IR partial spectra centered in the range from 1570 to 1830 cm^−1^ for Mat 2 and Nano 6 samples, respectively. Spectra presented in Figure 8A and Figure 9A correspond to purified Mat 2 and Nano 6, respectively, while spectra 8B and 9B are also of Mat 2 and Nano 6, respectively, but now after hydrocortisone release finished. Both spectra in each figure were normalized taking the absorbance of the most intense band as the unit. In this region, the spectral contributions due to the stretching vibration of carboxyl groups were resolved. It is evident the similarity between the peak spectral patterns shown on these figures. Thus, the Figure 8A and Figure 9A spectra show that the most intense band was detected at 1733 cm^−1^ in the spectrum of Mat 2 and at 1729 cm^−1^ in the spectrum of Nano 6, respectively. Stretching vibrations of carboxyl groups linked by inter-associated hydrogen bonding cause these spectral features [56]. For Mat 2, these hydrogen bonds are formed through an association between carbonyl groups, of BA and carboxyl groups of MAA, while for Nano 6, besides this association, the participation of free hydroxyl and carboxyl groups inserted on the walls of CNTs_r_ to form hydrogen bonds must be considered. In this last case, formation of hydrogen bonds with either carbonyl groups of BA or carboxyl groups of MAA is possible. This spectral behavior is induced by the pH (close to 7). Due to the fact that hydrocortisone has three hydroxyl groups and two carbonyl groups, it is expected that these functional groups can form hydrogen bonds with the corresponding functional groups of Mat 2 and Nano 6. After the hydrocortisone release ended, the complex balance of inter-associated or self-associated hydrogen bonds has changed. The Figure 8B and Figure 9B spectra show that now the most intense band was detected at 1700 cm^−1^ (Figure 8B) or at 1703 cm^−1^ (Figure 9B). Stretching vibrations of self-associated carboxyl groups linked by hydrogen bonds forming dimers that caused these bands [57]. This result means that now prevalence of hydrogen bonds by self-association over inter-association is occurring. Therefore, we consider that the behavior observed in the hydrocortisone release profile of Nano 6 is caused by the change in the balance of carboxyl groups (both in MAA units and in CNTs_r_) inter-associated with respect to self-associated by hydrogen bonds. In other words, the breaking of the hydrogen bonds, that kept it physically attached to the CNTs_r_, drives the release of hydrocortisone. This is due to the influence that pH has on the amount and nature of the hydrogen bonds formed by carboxylic acid groups.

For nanocomposites and MAA–BA copolymer (Mat 1) prepared from an initial formulation richest in BA, we did not obtain adequate specimens to make an appropriate hydrocortisone release study.

The other possibility for hydrocortisone release is the hydrolysis of anhydride groups (very sensitive to pH) [58] is ruled out, at least at pH 5, because there is no evidence of hydrolysis in the FT-IR spectra of the series of nanocomposites synthesized.

## 4. Conclusions

In this work, a smart drug delivery system was synthesized successfully. This system consists of nanocomposites of CNTs_o-OCl-MAA_–BA and CNTs_r-OCl-MAA_–BA that were synthesized by IEP. An analysis of FE-SEM and HR-TEM microscopies made it possible to observe that the CNTs synthesized by CVD have a structure of HMWCNTs. To achieve the synthesis of the aforementioned nanocomposites, HMWCNTs were previously functionalized chemically to attach the acyl chloride functionality onto their walls. This reactive functional group of the CNTs_o-OCl_ and CNTs_r-OCl_ reacted with the hydroxyl group of the methacrylic acid, creating an anhydride group, but keeping the C=C bond unchanged. Through radical polymerization of MAA and BA monomers on the substrate (CNTs_o-OCl-MAA_ or CNTs_r-OCl-MAA_), CNTs_o-OCl-MAA_–BA and CNTs_r-OCl-MAA_–BA nanocomposites were synthesized. Raman and FT-IR spectroscopies confirmed that the functionalization of the purified CNTs was carried out as shown in Scheme 1. XPS and ^13^C-NRM spectroscopies show that the MAA–BA chains are attached to CNTs_o-OCl-MAA_ or CNTs_r-OCl-MAA_ substrates. DSC thermograms evidenced that the co-monomer formulation and the amount of CNTs_o-OCl_ or CNTs_r-OCl_ used in the nanocomposites preparation determine the morphology of the synthesized nanocomposites. In general, nanocomposites synthesized with a higher amount of BA show two glass transition temperatures, while nanocomposites prepared with a higher amount of MAA show one glass transition temperature. There is no evidence that nanocomposites morphology influences their ability to release hydrocortisone. In the synthesized nanocomposites the CNTs_f_, incorporated in their structure, act as hydrocortisone reservoirs. Of the all nanocomposites studied, Nano 6 shows the best ability to release hydrocortisone. The hydrocortisone release of Nano 6 carried out at pH 5 starts after 73 h after having initiated the release test at room temperature. The hydrocortisone release profile that shows the Nano 6 is driven by a change in the inter-associated to self-associated hydrogen bonds balance. The polymeric pure matrix of the synthesized nanocomposites does not show a significant ability to release hydrocortisone. Therefore, the presence of the CNTs_r_ in the Nano 6 plays a crucial role in obtaining a smart drug delivery system.

Although the prepared nanocomposites in this work have a convenient ability to release hydrocortisone, the decisive role that the functionalized CNTs have in the drug delivery performance of nanocomposites urges the necessity to evaluate CNTs’ toxicity. The unresolved problem of toxicity of the CNTs must be overcome before their practical use can be explored. The toxicity of the CNTs is a risk that limit the possible use of the nanocomposites studied. This risk must be thoroughly evaluated to avoid undesirable health problems. Therefore, the possible impact of the studied nanocomposites on human health, by their use in a drug delivery system at pH 5, depends upon their evaluation as a potential threat to living tissues.

## Data Availability

The data presented in this study are available on request from the corresponding author.

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
