# Peer review of "Synthesis of Poly(methacrylic acid-co-butyl acrylate) Grafted onto Functionalized Carbon Nanotube Nanocomposites for Drug Delivery"

_polymers, 2021, doi:10.3390/polym13040533_

Round 1

Reviewer 1 Report

The authors present an interesting paper concerning the synthesis of PMAA-PBA copolymers grafted with CNT for drug delivery. Materials, methods and results are well presented, and the topic analyzed is of great interest. I believe that the paper can be published in "Polymers", but some questions must be clarified:

1 - Do you have any idea of the efficiency grafting process of the CNT? I think that due to the aggregation of the CNT, grafting could be not 100 % effective, and authors should discuss this point in the paper.

2 - DSC results were obtained during the first heating ramp. However, it is known that polymers could present "thermal history" that could affect the measurements, so it would be necessary to use a second heating ramp to verify the glass transition temperatures detected.

3 - Table 4 includes the glass transition temperatures measured, and the subsequent discussion is clear and large. However, authors should indicate the influence of these values, and the possible limitations derived, when using these copolymers in real drug delivery applications, for example, as inyectable hydrogels.

4 - Do the authors have measured the swelling behavior of these materials. In drug delivery applications, this is a key parameter that should be determined and discussed in the manuscript.

Author Response

Reviewer 1.

The authors present an interesting paper concerning the synthesis of PMAA-PBA copolymers grafted with CNT for drug delivery. Materials, methods and results are well presented, and the topic analyzed is of great interest. I believe that the paper can be published in "Polymers", but some questions must be clarified:

1 - Do you have any idea of the efficiency grafting process of the CNT? I think that due to the aggregation of the CNT, grafting could be not 100 % effective, and authors should discuss this point in the paper.

We agree with the reviewer. For the nanocomposites studied it is very unlikely to carry out a grafting reaction with 100% of efficiency. Several papers have been published related to CNTs based nanocomposites, in which TGA tests were used to evaluate the efficiency of the grafting reaction. Since last year, due to the COVID 19 pandemic, the use of the TGA equipment of the University of Guadalajara was restricted. Nowadays, it is not possible to use the TGA equipment. Thus, we cannot report the grafting reaction efficiency at this time. It is very unlikely that this situation can be remediated soon. As mention in the revised manuscript, achieving a complete grafting is ruled out.

2 - DSC results were obtained during the first heating ramp. However, it is known that polymers could present "thermal history" that could affect the measurements, so it would be necessary to use a second heating ramp to verify the glass transition temperatures detected.

We disagree with the reviewer. In the original manuscript (section 2.5) it was stated that the second DSC scans are reported. Therefore, glass transitions detected are correct.

3 - Table 4 includes the glass transition temperatures measured, and the subsequent discussion is clear and large. However, authors should indicate the influence of these values, and the possible limitations derived, when using these copolymers in real drug delivery applications, for example, as inyectable hydrogels.

As was explained in the manuscript, in general, nanocomposites synthesized from a higher content of BA in the initial formulation, two Tgs are shown. This means that domains rich in hydrophobic BA are formed in the polymer matrix of some nanocomposites. Other nanocomposites, in general, those prepared with 40wt.% MAA show only one Tg. This is typical of a random structure. In principle, it is expected that hydrocortisone would be captured by MAA units, included in polymer macromolecules of the nanocomposites or in their pure polymer matrix, via hydrogen bonds. In this sense, the morphology of the nanocomposites should have a strong influence on hydrocortisone delivery. But when the hydrocortisone delivery study was carried out, we found that only Nano 6 showed a significant rate release. Due to Nano 6 showing only one Tg its structure is random. Besides, the pure MAA-BA copolymers do not show an appreciable capacity to release hydrocortisone and, also showed one Tg. Therefore, we consider that the morphology of the nanocomposites or of the pure polymeric matrix is not the principal factor that has an influence on the hydrocortisone release. Thus, it must be the functionalized CNTs acting as hydrocortisone reservoirs, which are the cause of significant hydrocortisone release. Moreover, nanocomposites must contain sufficient amounts of functionalized CNTs, effectively inserted to achieve the desired release effect. Nano 5 prepared with CNTso, did not show an appreciable hydrocortisone release. Conversely, Nano 6 prepared with CNTsr showed a discrete hydrocortisone release. Besides, the results obtained by XPS and NMR spectroscopies evidence a higher formation of anhydride groups in Nano 6 than in Nano 5. Thus, it is reasonable to consider that the content of functionalized CNTs in Nano 6 must be larger than in Nano 5. Consequently, Nano 6 has larger reservoirs and a better capacity to release hydrocortisone.

With respect to the possibility of using the studied nanocomposites in real form, for example, as hydrogels, there is an unresolved problem that should be overcome first. To our knowledge, there are still contradictory reports concerning the toxicity of CNTs. Some authors report that functionalized CNTs are less toxic than pure CNTs. Accordingly, we designed nanocomposites containing functionalized CNTs, with the hope that they would have the capacity to release drugs without causing health problems. But to clarify this point, it is necessary to conduct further studies.

4 - Do the authors have measured the swelling behavior of these materials. In drug delivery applications, this is a key parameter that should be determined and discussed in the manuscript.

As the reviewer stated; the swelling is a significant factor in drug delivery applications. We measured the swelling degree of studied nanocomposites and their pure polymer matrix at room temperature. For this, tablets of both types of mentioned materials were dried to a constant weight (Wd). The dried tablets were placed in a vial with 10 mL of potassium biphthalate buffer, pH 5, for 24 h. The swollen tablets were then taken out of the vial at certain times, wiped with a filter paper, and weighed (Ww). The swelling degree (Sw) was calculated according to equation (1):

Sw=(Ww-Wd)/Wd                                   (1)

Where Wd is the weight of the dried tablet and Ww is the weight of the swollen tablet.

After 48 h, Sw cannot be obtained. These results are the consequence of two factors: (i) the nanocomposites studied, and their matrix polymeric are mainly hydrophobic and, (ii) at pH 5, the units of MAA inserted in the nanocomposites or in their polymer matrices are self-associated by hydrogen bonds. Thus, they are unavailable to create physical interactions with water. Therefore, their capacity to uptake water is null. In the revised manuscript we added a brief discussion about this, both in the section “2.5 Characterization of MAA-BA copolymers and their CNTso-OCl-MAA-BA and CNTsr-OCl-MAA-BA nanocomposites”, as well as in Section 3. “Results and Discussion”.

Reviewer 2 Report

The manuscript "SYNTHESIS OF POLY(METHACRYLIC ACID-co-BUTYL ACRYLATE)-GRAFTED TO FUNCTIONALIZED CARBON NANOTUBES NANOCOMPOSITES FOR DRUG DELIVERY" is having interesting findings and solves an interesting topic.  However a major revision is required before publication in MDPI Polymers journal.

My advice, comments and recommendations are listed below:

The language of the manuscript should be improved so that it is easy to read. You need to correct the grammar. Please go through the entire manuscript and shorten and correct some sentences.

I recommend clarifying and improve presentation of abstract so that it is clear to the reader what this is all about. Include your recommendations and future prospects. It is necessary to extend the abstract with the most significant results.

Line 29: ,,13C-RMN,, you probably mean ,,13C-NMR, if so, correct it throughout text.

Please be sure that your manuscript thoroughly establishes how this work is fundamentally novel. Specific comparisons should be made to previously published materials that have a similar purpose. Please present a strong case for how this work is a major advance. This needs to be done in the manuscript itself, not just in the response to review comments.

Please be sure that your abstract and your Conclusions section not only summarize the key findings of your work but also explain the specific ways in which this work fundamentally advances the field relative to prior literature.

The significance of this study should be more emphasize in the introduction.

Line 44: Take a look these two very important papers, which can help you. https://www.sciencedirect.com/science/article/abs/pii/S0014305707006337

and https://www.sciencedirect.com/science/article/pii/S1818087620300659

Line 101, section Poly(methacrylic acid) (PMAA): This statement support this very important paper and therefore, authors are encouraged to add it as a reference to this place. https://www.mdpi.com/2073-4360/12/3/708

Line 139: ,,Alfa Aeser,, change to ,,Alfa Aesar,,

Line 140: Remove percentages of purity from these gases. It's confusing. 

Line 160: Please explain this chapter more clearly.

Line 189: Check the correct naming and use of SI units in whole manuscript.

Line 234: Improve the wording of the title as well as the interpretation of Table 1. For example, start like this ,, Content of pure MAA-BA copolymers, CNTso-OCl-MAA-BA and CNTsr-OCl-MAA-BA nanocomposites used for the preparation. You do not need to specify wt.% in the table name, since it is already in the table 1 itself.

Line 242: I like this scheme, but please check it to see if it's whole correct.

Line 251: ,,RMN,, change to ,,NMR,,

Line 253: In this section, edit and shorten some sentences to make them easier to understand and clear.

Line 260: Add the country of origin in whole manuscript of each device on which the experiments were performed.

Line 275: ,,Samples of CNTsp, functionalized CNTs, CNTso-OCl-MAA-BA and CNTsr-OCl-MAA-BA nanocomposites and the MAA-BA copolymers were analyzed by Fourier-transform infrared spectroscopy (FT-IR) with a spectrophotometer,, change to ,, Structural characterization of CNTsp, functionalized CNTs, CNTso-OCl-MAA-BA and CNTsr-OCl-MAA-BA nanocomposites and the MAA-BA copolymers was performed by Fourier-transform infrared spectroscopy (FT-IR spectrophotometer, Spectrum One of Perkin Elmer).,, then add the country of origin of the device in parentheses.

Line 278: Add the sample / KBr ratio to the manuscript. Usually a ratio of 1: 200 is used, better said ratio 1 mg of sample and 200 mg of KBr.

Line 280: ,,40 scans,, this figure is not reliable usually 32, 64 or 128 scans are used. Please verify it. 

Line 331: Insert pictures 1A and 1B next to each other and name them together. 

Line 392: Distinguish Raman spectra in color. Distinguish color Raman spectra and number the most important peaks.

Line 433, section C-H bonds of methyl groups: This statement support this very important paper and therefore it must be add to this place. See also other important vibration modes in this paper. https://www.sciencedirect.com/science/article/abs/pii/S0169131719301413

Line 439: Distinguish the IR spectra in color.

Line 441: ,,1H spectra of Mat 1 and Mat 2 were obtained (not shown here),, and where are they then? You can't write that like this. Either don't mention it or add it to the manuscript or add to the supporting information.

Line 470: Figure 4 is faintly visible, improve its sharpness.

Line 472: Try to write this part more clearly.

Line 498: Figure 5 has great results but is difficult to see. Improve the quality of the figure 5.

Line 516: Improve the interpretation of Table 3.

Line 556: Please verify the accuracy of these results. The y-axis needs to be corrected and added (Heat flow/mW g-1). I would appreciate adding DTG results.

Line 566: Edit Table 4. Improve the presentation of the results in this table.

Line 590: Figure 7 is faintly visible. Improve it.

Line 622 and 624: These spectra need to be color-coded to better see the differences.

Line 633: Extend the conclusions with all your most important findings. Indicate the possible risks of such research. Add your recommendations for future research.

Make sure the references are added correctly according to the journal's instructions.

Author Response

Reviewer 2.

The manuscript "SYNTHESIS OF POLY(METHACRYLIC ACID-co-BUTYL ACRYLATE)-GRAFTED TO FUNCTIONALIZED CARBON NANOTUBES NANOCOMPOSITES FOR DRUG DELIVERY" is having interesting findings and solves an interesting topic.  However a major revision is required before publication in MDPI Polymers journal.

My advice, comments and recommendations are listed below:

The language of the manuscript should be improved so that it is easy to read. You need to correct the grammar. Please go through the entire manuscript and shorten and correct some sentences.

The grammar in all manuscript was reviewed. Some sentences were shortened to improve the clarity of the manuscript

I recommend clarifying and improve presentation of abstract so that it is clear to the reader what this is all about. Include your recommendations and future prospects. It is necessary to extend the abstract with the most significant results.

The abstract was improved and it was limited to 180 words to meet the mandatory format of the Polymers journal.

Line 29: ,,13C-RMN,, you probably mean ,,13C-NMR, if so, correct it throughout text.

We agree with the reviewer. In the revised manuscript, RMN word was changed to the NMR word.

Please be sure that your manuscript thoroughly establishes how this work is fundamentally novel. Specific comparisons should be made to previously published materials that have a similar purpose. Please present a strong case for how this work is a major advance. This needs to be done in the manuscript itself, not just in the response to review comments.

In the revised manuscript, the last paragraph of the Introduction was rewritten. This was done in order to present the relevant achievements of this work, as well as to show their novelty. Also, in the Introduction section, a new paragraph was added (located before describing some characteristics of the hydrocortisone) to broader the background for this work. This was made for sake of improving the clarity of the fundamentals over was developed our work. Two references cited in this paragraph (references 24 and 25) are now included, as a strong solid base from which are presented the advances of our work.

Please be sure that your abstract and your Conclusions section not only summarize the key findings of your work but also explain the specific ways in which this work fundamentally advances the field relative to prior literature.

Both Abstract and the Conclusions were modified in order to summarize the key findings obtained in the present work.

The significance of this study should be more emphasize in the introduction.

We agree with the reviewer. We added two new paragraphs in the Introduction section of the revised manuscript. Another paragraph (which is presented information about the BA) was moved just after the paragraph about the PMAA. Also, eight new references were included and the last paragraph was rewritten. These changes emphasize the significance of this study.

Line 44: Take a look these two very important papers, which can help you. https://www.sciencedirect.com/science/article/abs/pii/S0014305707006337, and https://www.sciencedirect.com/science/article/pii/S1818087620300659

The two papers suggested by the reviewer were taken into account and cited in the Introduction of the revised manuscript.

Line 101, section Poly(methacrylic acid) (PMAA): This statement support this very important paper and therefore, authors are encouraged to add it as a reference to this place. https://www.mdpi.com/2073-4360/12/3/708

The reference suggested by the reviewer was added to the revised manuscript.

Line 139: ,,Alfa Aeser,, change to ,,Alfa Aesar,,

This mistake was corrected, Aeser was changed by Aesar.

Line 140: Remove percentages of purity from these gases. It's confusing. 

The percentages of purity of both gases: nitrogen and argon were removed.

Line 160: Please explain this chapter more clearly.

This chapter was rewritten to improve its clarity.

Line 189: Check the correct naming and use of SI units in whole manuscript.

In the revised manuscript, all units are written in SI units, with the exception of the units for time; for which we use hours and minutes are used instead of seconds because we consider that for this way, the manuscript is more clear and easier to read.

Line 234: Improve the wording of the title as well as the interpretation of Table 1. For example, start like this ,, Content of pure MAA-BA copolymers, CNTso-OCl-MAA-BA and CNTsr-OCl-MAA-BA nanocomposites used for the preparation. You do not need to specify wt.% in the table name, since it is already in the table 1 itself.

The wording of the Table 1 title was improved. Also, the interpretation of the data listed in Table 1 was rewritten to increase their clarity.

Line 242: I like this scheme, but please check it to see if it's whole correct.

We revised scheme 1 and the subscripts of the co-monomeric units were removed to eliminate a confusing interpretation. In the revised manuscript, a corrected scheme 1 was included.

Line 251: ,,RMN,, change to ,,NMR,,

We agree. In the revised manuscript, the RMN word was changed to the NMR word.

Line 253: In this section, edit and shorten some sentences to make them easier to understand and clear.

We agree with the reviewer. The section mentioned in this request was rewritten to improve its clarity.

Line 260: Add the country of origin in whole manuscript of each device on which the experiments were performed.

We included the country of origin of each device used in this work.

Line 275: ,,Samples of CNTsp, functionalized CNTs, CNTso-OCl-MAA-BA and CNTsr-OCl-MAA-BA nanocomposites and the MAA-BA copolymers were analyzed by Fourier-transform infrared spectroscopy (FT-IR) with a spectrophotometer,, change to ,, Structural characterization of CNTsp, functionalized CNTs, CNTso-OCl-MAA-BA and CNTsr-OCl-MAA-BA nanocomposites and the MAA-BA copolymers was performed by Fourier-transform infrared spectroscopy (FT-IR spectrophotometer, Spectrum One of Perkin Elmer).,, then add the country of origin of the device in parentheses.

This suggestion proposed by the reviewer was accepted and included in the revised manuscript.

Line 278: Add the sample / KBr ratio to the manuscript. Usually a ratio of 1: 200 is used, better said ratio 1 mg of sample and 200 mg of KBr.

In the revised manuscript the ratio of sample and KBr used was included.

Line 280: ,,40 scans,, this figure is not reliable usually 32, 64 or 128 scans are used. Please verify it. 

We reviewed and found that the reviewer’s observation is correct. The true number of scans was 64. In the revised manuscript, this mistake was corrected.

Line 331: Insert pictures 1A and 1B next to each other and name them together. 

In the revised manuscript both figures 1A and 1B were inserted next to each other and only one legend was written.

Line 392: Distinguish Raman spectra in color. Distinguish color Raman spectra and number the most important peaks.

Raman spectra were distinguished by color. Moreover, the number of the highlighted peaks was added.

Line 433, section C-H bonds of methyl groups: This statement support this very important paper and therefore it must be add to this place. See also other important vibration modes in this paper. https://www.sciencedirect.com/science/article/abs/pii/S0169131719301413

After reading the paper suggested by the reviewer, we added it to the revised manuscript.

Line 439: Distinguish the IR spectra in color.

IR spectra were now drawn by color.

Line 441: ,,1H spectra of Mat 1 and Mat 2 were obtained (not shown here),, and where are they then? You can't write that like this. Either don't mention it or add it to the manuscript or add to the supporting information.

We decided that 1H spectra of Mat 1 and Mat 2 should be shown in the supporting information. In the revised manuscript, we wrote that both 1H spectra are shown in the supporting information section.

Line 470: Figure 4 is faintly visible, improve its sharpness.

In the revised manuscript, the sharpness of figure 4 was increased.

Line 472: Try to write this part more clearly.

In the revised manuscript, line 472 was rewritten to improve its clarity.

Line 498: Figure 5 has great results but is difficult to see. Improve the quality of the figure 5.

In the revised manuscript, the quality of figure 5 was improved.

Line 516: Improve the interpretation of Table 3.

The interpretation of Table 3 was improved.

Line 556: Please verify the accuracy of these results. The y-axis needs to be corrected and added (Heat flow/mW g-1). I would appreciate adding DTG results.

Although it is evident that some of the Tgs reported are subtle, the accuracy of the values reported was verified. The y-axis of Figure 6 was changed, adding the legend indicated by the reviewer. With respect to adding a DTG analysis, we must mention that because of the COVID-19 pandemic, the access to laboratories of the University of Guadalajara has been canceled since January 12, 2021. For the moment, we cannot perform tests by DTG. We do not know when it will be possible to conduct said studies. We apologize for not presenting a DTG analysis for the submitted manuscript. However, we consider that the results of DTG analysis would not change the thermal behavior of the nanocomposites observed by the DSC technique.

Line 566: Edit Table 4. Improve the presentation of the results in this table.

Table 4 was edited. In the revised manuscript, a third column was added to separate the two values reported in one column presented in the original manuscript.

Line 590: Figure 7 is faintly visible. Improve it.

In the revised manuscript the released hydrocortisone curves were shown in color, improving the clarity of figure 7.

Line 622 and 624: These spectra need to be color-coded to better see the differences.

Figures 8 and 9 were distinguished by different colors.

Line 633: Extend the conclusions with all your most important findings. Indicate the possible risks of such research. Add your recommendations for future research.

The conclusions were extended. In addition, risks and recommendations were added.

Make sure the references are added correctly according to the journal's instructions.

References were checked and the DOI number was added for available references.

Round 2

Reviewer 2 Report

The manuscript ,,Synthesis of Poly(methacrylic acid-co-butyl acrylate)-Grafted to Functionalized Carbon Nanotubes Nanocomposites for Drug Delivery,, has been significantly improved. The authors have made great efforts.

The manuscript can now be accepted in present form as a publication in the journal Polymers.